# Detection and Simultaneous Differentiation of Three Co-infected Viruses in *Zanthoxylum armatum*

**DOI:** 10.3390/plants11091242

**Published:** 2022-05-05

**Authors:** Zhenfei Dong, Xiaoli Zhao, Junjie Liu, Binhui Zhan, Shifang Li

**Affiliations:** 1State Key Laboratory for Biology of Plant Diseases and Insect Pests, Institute of Plant Protection, Chinese Academy of Agricultural Sciences, Beijing 100193, China; dongzhenfeiqaz@163.com; 2Department of Fruit Science, College of Horticulture, China Agricultural University, Beijing 100107, China; zhaoxiaolicau@163.com (X.Z.); 15806361586@163.com (J.L.)

**Keywords:** multiplex RT-PCR, *Zanthoxylum armatum*, GSPNeV, GSPIV, GSPEV, ITS2

## Abstract

Green Sichuan pepper (*Zanthoxylum armatum*) is an important economic fruit crop, which is widely planted in the southwest region of China. Recently, a serious disease, namely flower yellowing disease (FYD), broke out, and the virus of green Sichuan pepper nepovirus (GSPNeV) was identified to be highly correlated with the viral symptoms. Meanwhile, green Sichuan pepper idaeovirus (GSPIV) and green Sichuan pepper enamovirus (GSPEV) were also common viruses infecting green pepper. In our research, specific primers were designed according to the reported sequences of the three viruses, and a multiplex reverse transcription-polymerase chain reaction (RT-PCR) method for the simultaneous detection of GSPNeV, GSPIV, and GSPEV was established. The annealing temperature, extension time, and cycle number affecting the multiplex RT-PCR reaction were adjusted and optimized. Sensitivity analysis showed that the system could detect the three viruses simultaneously from the complementary deoxyribonucleic acid (cDNA) samples diluted by 10^−3^. The results of the ten samples detected by the multiplex RT-PCR system were consistent with the results of a single PCR, indicating that the method can be successfully used for rapid detection of field samples.

## 1. Introduction

As one of the eight major cuisines in China, Sichuan cuisine is loved by people all over the world, and green Sichuan pepper (*Zanthoxylum armatum*) is one of the most important spices [1]. Green Sichuan pepper is a kind of deciduous, spiny shrub or small tree in the genus *Zanthoxylum* and family Rutaceae, which has a long history of cultivation in China [2]. In addition to being used as a spice, it also has applications in pharmaceutical and cosmetics industries [2,3]. Green Sichuan pepper is mainly distributed in tropical and subtropical regions, and widely planted in Asian countries, especially in the southwest regions in China [4]. Particularly, green Sichuan pepper is of great economic value, for example, with approximately 530,000 acres of planting area and a sale value of over half a billion US dollars alone in Jiangjin District, Chongqing Municipality, China, in 2018 [5]. Recently, a virus-like disease, the flower yellowing disease (FYD), has emerged, mainly in the planting areas in China, and has gradually become the main restriction factor in the development of green Sichuan pepper production [6]. FYD can lead to a severe disorder of the floral organs, with the symptoms of pistil abortion and stamen yellowing, usually with yellowing and stunting of the foliage, and can ultimately be destructive to fructification of the diseased plants (Figure 1) [5]. This disease usually occurred firstly on a few branches and progressively spread to the whole plant, causing an irreversible decline and eventually death in the trees, with huge economic losses [7]. 

The technology of high-throughput sequencing (HTS) was applied to investigate the pathogens associated with FYD and revealed the virome existing in the FYD-affected green Sichuan pepper. Several viruses were identified, as follows: green Sichuan pepper nepovirus (GSPNeV) (Family *Secoviridae*, Genus *Nepovirus*), green Sichuan pepper nepovirus large satellite (satGSPNeV), green Sichuan pepper idaeovirus (GSPIdV or GSPIV) (Family unassigned, Genus *Idaeovirus*), green Sichuan pepper enamovirus (GSPEV) (Family *Luteoviridae*, Genus *Enamovirus*), green Sichuan pepper nucleorhabdovirus (GSPNuV) (Family *Rhabdoviridae*, Genus *Nucleorhabdovirus*), green Sichuan pepper vein-clearing-associated virus (GSPVCaV) (Family *Caulimoviridae*, Genus *Badnavirus*), and green Sichuan pepper ilarvirus (GSPIlV) (Family *Bromoviridae*, Genus *Ilarvirus*) [5,7,8]. Among them, GSPNeV and satGSPNeV were major factors associated with the FYD-infected trees [5,7]. Moreover, HTS analysis showed that reads of GSPNeV were abundant in symptomatic branches but deficient in asymptomatic branches of the same tree, which indicated the high relationship between GSPNeV and FYD. In addition, HTS also revealed that the reads of GSPIV, GSPEV, and GSPIlV were of moderate abundance in symptomatic and asymptomatic trees, even with minor effects on FYD [5]. 

The diagnosis of viruses is important in the disease management [9]. Serology-based methods (e.g., enzyme-linked immunosorbent assay, western) and nucleic acid-based methods (e.g., polymerase chain reaction (PCR)) are two kinds of important and popular diagnostic technologies [10,11]. Multiplex reverse transcription (RT)-PCR is one of the nucleic acid-based methods which is designed for specific detection of several viruses within one tube at a time, providing a quick, reliable, and cost-effective method [12]. In addition to the primer pairs for virus detection, the primer pair for the detection of plant endogenous fragments was also used as an indicator to validate the quality of the plant extracts and the process of reverse transcription-polymerase chain reaction (RT-PCR) [13]. 

In this study, a two-step multiplex RT-PCR method was developed for the simultaneous detection of GSPNeV, GSPIV, and GSPEV in green Sichuan pepper. The internal transcribed spacer 2 (ITS2) was used as an internal control to check the ribonucleic acid (RNA) quality and the effectiveness of RT-PCR.

## 2. Results

### 2.1. Validation of Specific Primers

The genome sequences of different GSPNeV, GSPIV, and GSPEV isolates (including 12 GSPNeV isolates, 14 GSPIV isolates, and 15 GSPEV isolates reported so far) were downloaded from the NCBI database. The conserved sequences were used for primer design. The sequence information of ITS2 (GenBank Accession No. KX192307.1) was downloaded from the NCBI database and used for internal control gene primer design. The primers were 16–20 bp in length, with no more than two single-nucleotide polymorphisms present in the last nucleotide position at the 3′-terminal ends. The primer names, primer sequences, and the expected size of amplicons are listed in Table 1.

To evaluate the primer specificity, the simplex PCR was conducted. The 15 μl reaction containing 2× Taq Mix (Sangon Biotech, Shanghai, China) 7.5 μl, 1 μl of complementary deoxyribonucleic acid (cDNA), 0.4 μl of forward primer (10 μmol/L), and 0.4 μl of reverse primer (10 μmol/L) was set up for each virus (GSPNeV, GSPIV, and GSPEV). The PCR was performed in a thermal cycler (BIO-RAD, Hercules, CA, USA) using the following procedure: denaturation at 94 °C for 3 min, followed by 35 cycles at 94 °C for 30 s, 54 °C for 30 s, and 72 °C for 60 s, which was then followed by a final extension at 72 °C for 5 min. The PCR products were analyzed on a 1.5% agarose gel and visualized under ultraviolet light. Specific PCR amplification products of the expected sizes (1039 bp for GSPNeV, 730 bp for GSPIV, and 580 bp for GSPEV) were obtained from the FYD samples, including the above three viruses (Figure 2). No amplification products were obtained from the healthy plants (Figure 2). 

For multiplex RT-PCR, the 15 μl reaction mixture included 7.5 μl of 2× Taq Mix (Sangon Biotech, Shanghai, China), 1 μl of cDNA, and 3.2 μl of multiplex primer mix (containing 0.4 μl each of forward and reverse primer for GSPNeV, GSPIV, GSPEV, and ITS2 from 10 mΜ stock). The PCR amplification was performed as described in simplex PCR. After electrophoresis, the PCR products were visualized under UV light and the specificities of the primers for the target viruses were further validated by subsequent cloning and sequencing. The sequences of amplified fragments were compared with the published sequences at GenBank. The results showed that the amplified fragment of GSPNeV shared 93.84–99.81% similarity with different GSPNeV isolates, the amplified fragment of GSPIV shared 92.23–100% similarity with different GSPIV isolates, and the amplified fragment of GSPEV shared 88.43–98.96% similarity with different GSPEV isolates. The amplified fragment of ITS2 shared 94.69% similarity with the *Zanthoxylum bungeanum* isolate LHS3 ITS2, partial sequence (GenBank Accession No. KX192307.1).

### 2.2. Optimization of Multiplex RT-PCR

For the optimization of multiplex RT-PCR, the parameters of primer concentration, annealing temperature, extension time, and recycles were evaluated. The concentrations of GSPNeV, GSPIV, GSPEV, and the internal control ITS2 specific primers were adjusted from 0.1 to 0.6 μl. The final concentrations of each specific forward and reverse primer were 0.4 μl for GSPNeV, 0.6 μl for GSPIV, 0.1 μl for GSPEV, and 0.2 μl for ITS2, respectively (data not shown). The extension times of 30, 60, and 90 s were tested, and the best results were obtained at 60 or 90 s (Figure 3a). The annealing temperatures at 52, 54, and 56 °C were tested and the results did not show significant changes (Figure 3b). The amplification cycles were set at 25, 30, and 35, and the band intensity was significantly improved above 30 cycles (Figure 3c).

### 2.3. Sensitivity Detection of Multiplex RT-PCR

To evaluate the sensitivity of multiplex RT-PCR, a series of 10-fold dilutions (10^−^^1^ to 10^−7^) of cDNA from 1 μg of total RNA of the positive sample were used for detection. GSPNeV, GSPIV, GSPEV, and ITS2 showed positive results at the dilution of 10^−3^, while ITS2 was not visible and bands of GSPNeV, GSPIV, and GSPEV were weak at the dilution of 10^−4^. Above all, the detection limit of the three viruses and the internal control ITS2 was at a 10^−3^ dilution (Figure 4).

### 2.4. Detection of Field Samples by the Optimized Multiplex RT-PCR

To validate the reliability and efficiency of the optimized multiplex RT-PCR, the field green Sichuan pepper samples infected by different viruses were tested. In these ten samples, all samples were ITS2- and GSPEV-positive, nine samples were GSPIV-positive, and four samples were GSPNeV-positive (Figure 5). Moreover, four samples showed mixed infection by GSPNeV, GSPIV, and GSPEV, five samples were co-infected by GSPIV and GSPEV, and only one sample was solely infected by GSPEV (Figure 5). The above results were consistent with those of simplex PCR tests, indicating that the established multiplex RT-PCR can be applied for field samples’ detection.

## 3. Discussion

Viral diseases are difficult to control with chemical methods, and prevention has a pivotal position in disease-controlling strategies, especially for perennial woody plants. The quick and reliable detection of viruses is important in prevention [14]. FYD was firstly identified in green Sichuan pepper in 2018, which caused serious economic losses [5]. In green Sichuan pepper plants with FYD, the phenomenon of mixed infection by different viruses is prevalent, although some viruses contribute little to the symptoms, but the potential effects cannot be ruled out [5]. In this paper, we selected GSPNeV, GSPIV, and GSPEV as the detection targets because GSPNeV was highly associated with FYD, and GSPIV and GSPEV were present in green Sichuan pepper with relatively high detection rates according to the previous disease survey [5,7].

Nucleic acid-based detection methods and serology-based methods are two kinds of commonly used detection methods. The serology-based methods are totally dependent on the specific antibodies, which are time-consuming and difficult to prepare, especially for the newly discovered viruses without knowing their biological characteristics. Nucleic acid-based detection methods, which are represented by PCR, only require partial genomic information. Recently, some newly developed nucleic acid-based detection methods, such as loop-mediated isothermal amplification system and recombinase polymerase amplification, were also successfully used for fast and sensitive detection of different viruses, while the high testing costs restricted their extensive application in routine detection [15,16]. RT-PCR detection is by far the most widely used method, however when it comes to detecting several different targets, it takes too long one-by-one. The multiplex RT-PCR assay was developed to solve this problem with high efficiency. In this paper, the multiplex RT-PCR was established to simultaneously detect three viruses (GSPNeV, GSPIV, and GSPEV) and one internal control (ITS2) in one tube, with a sensitivity similar to that of the conventional simplex RT-PCR based on the agarose gel electrophoresis. Moreover, the established multiplex RT-PCR assay was applied in the test of field-collected samples, which indicated that the system can be successfully used to detect three viruses from different genus in green Sichuan pepper with high speed, sensitivity, and accuracy.

## 4. Materials and Methods

### 4.1. Plant Materials

Samples used in the study were collected from a young green Sichuan pepper orchard in Jiangjin District, Chongqing Municipality, China.

### 4.2. Extraction of Total RNA and RT-PCR

Plant RNA was extracted with the cetyltrimethylammonium bromide (CTAB) method. Briefly, the leaf tissue (0.1 g) was fully ground in liquid nitrogen and extracted with 1 mL of 2% CTAB buffer (2% CTAB, 4% PVP-40, 100 mM Tris-HCl (pH 8.0), 25 mM EDTA (pH 8.0), 2 M NaCl, and 1% β-mercaptoethanol) (all chemicals were from Sigma-Aldrich, Sant Louis, MO, USA), vortexed, and incubated at 65 °C for 20 min. After centrifuging at 12,000 *g* for 15 min, the supernatant was transferred to a fresh centrifuge tube, added with an equal volume of chloroform, vortexed, and incubated at room temperature for 3 min. After centrifuging at 12,000 *g* for 15 min, the supernatant was transferred and mixed with 1/3 of the supernatant volume of LiCl (8 M) to precipitate RNA. The mixture was kept at −80 °C for 30 min or −20 °C overnight and centrifuged at 12,000 *g* for 20 min. The pellet was washed with 75% ethanol twice and dissolved in RNase-free water. The RNA concentration and quality were estimated using a Nanodrop 2000 spectrophotometer (Thermo Fisher Scientific, Waltham, MA, USA) and electrophoresis. For cDNA synthesis, RT was performed in a 20 µl reaction volume containing 1 µg of total RNA from leaf tissue, 4 µL of 5× M-MLV Reaction Buffer, 1 µL of random primer, 2 µL of 2.5 mM dNTPs, 1 µL of RNase Inhibitor (40 units/µL), and 1 µL of M-MLV reverse transcriptase (100 unit/µL). The reaction tubes were incubated at 42 °C for 60 min, followed by 5 min at 70 °C. 

### 4.3. PCR Amplification

For simplex PCR, the 15 μl reaction containing 7.5 μl of 2× Taq Mix, 1 μl of cDNA, 0.4 μl of forward primer (10 μmol/L), and 0.4 μl of reverse primer (10 μmol/L) was set up for each virus. For multiplex RT-PCR, the 15 μl reaction mixture included 7.5 μl of 2× Taq Mix, 1 μl of cDNA, and 3.2 μl of multiplex primer mix (containing 0.4 μl each of forward and reverse primer for GSPNeV, GSPIV, GSPEV, and ITS2 from 10 mΜ stock). The PCR amplification was performed as follows: denaturation at 94 °C for 3 min, followed by 35 cycles at 94 °C for 30 s, 54 °C for 30 s, and 72 °C for 60 s, followed by a final extension at 72 °C for 5 min and stored at 4 °C.

## 5. Conclusions

This study described a multiplex RT-PCR method for the simultaneous detection of three viruses from different genus and one internal control (ITS2) in green Sichuan pepper plants. The multiplex RT-PCR method is time-saving and cost-effective, and could be used for the specific detection of several viruses at once. In our study the positive results of viruses and ITS2 were obtained when the templated cDNA product were diluted until 10^−3^. The multiplex RT-PCR assay has been successfully utilized for the detection of field-collected samples. To our best knowledge, this study is the first report on utilizing multiplex RT-PCR for simultaneous detection of GSPNeV, GSPIV, and GSPEV in green Sichuan pepper.

## Figures and Tables

**Figure 1 plants-11-01242-f001:**
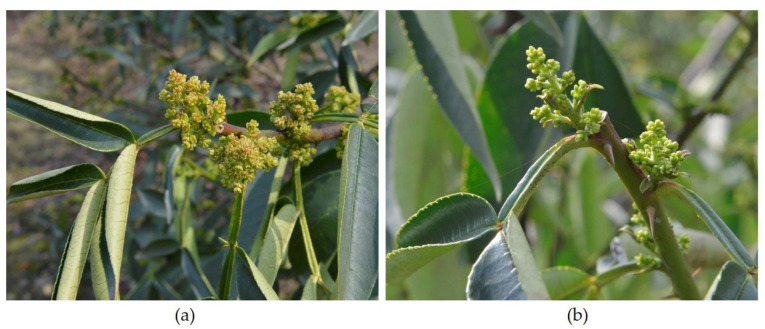
The symptoms of green Sichuan pepper with flower yellowing disease (**a**) and the healthy plant (**b**).

**Figure 2 plants-11-01242-f002:**
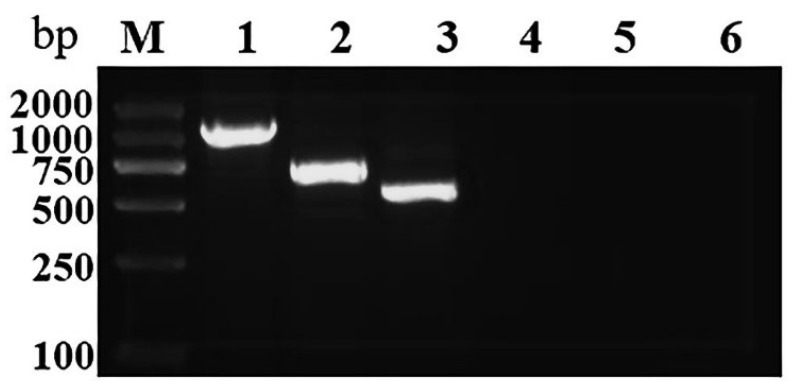
Detection of three viruses by simplex PCR. Lane M: Trans2K DNA marker. Lanes 1–3: FYD sample (positive sample) with specific primers for green Sichuan pepper nepovirus (GSPNeV), green Sichuan pepper idaeovirus (GSPIV), and green Sichuan pepper enamovirus (GSPEV). Lanes 4–6: healthy sample (negative sample) with specific primers for GSPNeV, GSPIV, and GSPEV.

**Figure 3 plants-11-01242-f003:**
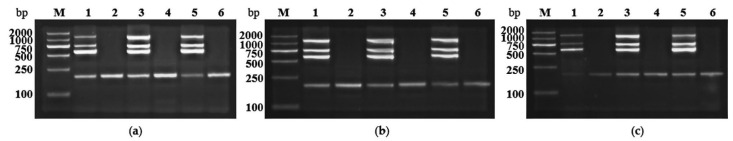
Establishment of multiplex RT-PCR assay for detection of three viruses. (**a**) Optimization of RT-PCR extension time. Lane M: Trans2K DNA marker. Lanes 1, 3, and 5: 30, 60, and 90 s for positive sample. Lanes 2, 4, and 6: 30, 60, and 90 s for negative sample. (**b**) Optimization of RT-PCR annealing temperature. Lane M: Trans2K DNA marker. Lanes 1, 3, and 5: 52, 54, and 56 °C for positive sample. Lanes 2, 4, and 6: 52, 54, and 56 °C for negative sample. (**c**) Optimization of RT-PCR amplification cycles. Lane M: Trans2K DNA marker. Lanes 1, 3, and 5: 25, 30, and 35 for positive sample. Lanes 2, 4, and 6: 25, 30, and 35 for negative sample.

**Figure 4 plants-11-01242-f004:**
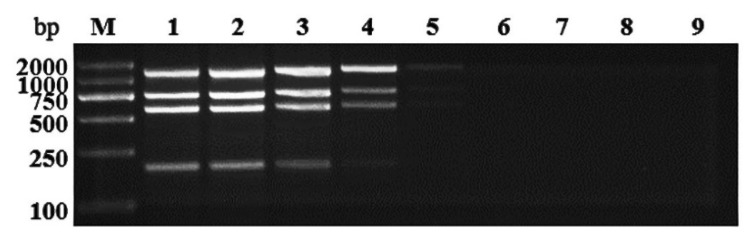
Sensitivity detection of multiplex RT-PCR using diluted cDNA. Lane M: Trans2K DNA marker. Lanes 1–8: a series of dilutions of cDNA from 1 μg of total RNA from the positive sample, including 10^0^, 10^−1^, 10^−2^, 10^−3^, 10^−4^, 10^−5^, 10^−6^, and 10^−7^. Lane 9: non-template control (NTC).

**Figure 5 plants-11-01242-f005:**
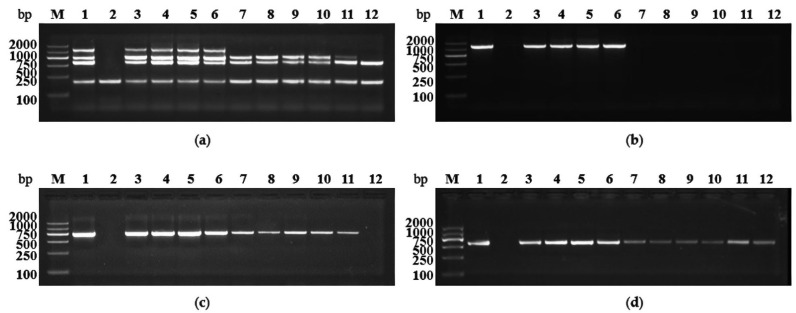
Detection results of field green Sichuan pepper samples by multiplex and simplex RT-PCR. (**a**) Detection of GSPNeV, GSPIV, and GSPEV simultaneously by multiplex RT-PCR. (**b**) Detection of GSPNeV by simplex RT-PCR. (**c**) Detection of GSPIV by simplex RT-PCR. (**d**) Detection of GSPEV by simplex RT-PCR. Lane M: Trans2K DNA marker, Lane 1: positive sample, Lane 2: negative sample, Lanes 3–12: field-collected green Sichuan pepper samples from Jiangjin District.

**Table 1 plants-11-01242-t001:** List of primers used in the multiplex detection of green Sichuan pepper nepovirus (GSPNeV), green Sichuan pepper idaeovirus (GSPIV), green Sichuan pepper enamovirus (GSPEV) and internal transcribed spacer 2 (ITS2).

Viruses or Gene	Primers	Primer Sequences (5′-3′)	Fragment Length (bp)
GSPNeV	GSPNeV-5F	TGTCACTGAAGGAGCGGAT	1039
	GSPNeV-3R	ATCACCCATCTTAGCGACG	
GSPIV	GSPIV-5F	GATTAGGGCACACAGGAGTC	730
	GSPIV-3R	GCTATGGCTTCTTCACGAA	
GSPEV	GSPEV-5F	CTGGGTTATGGCAAACATC	580
	GSPEV-3R	AGGAGCCTCAGGAAGAATCT	
ITS2	ITS2-5F	CGCATCGTTGCCCCAC	224
	ITS2-3R	CGATGCGAGCGCTGCTT	

## Data Availability

Not applicable.

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
