# Peer review of "Detection and Simultaneous Differentiation of Three Co-infected Viruses in Zanthoxylum armatum"

_plants, 2022, doi:10.3390/plants11091242_

Round 1

Reviewer 1 Report

The manuscript titled with" Simultaneous detection and differentiation of three co-infected viruses in Zanthoxylum armatum" considered one of the good research dealing with discovering a new and easy methods for virus detection in short time. But I have some comments in the research article:

First the authors did not make purification for any virus of the three studied viruses in the infected plant tissues and examined this under Transmission Electron Microscope to be quite sure that they are working on baseline.

-They did not mentioned how many genomes for different isolates of the same viruses were used to construct its specific primers. Because this step will ensure the accuracy of the designed used primers.

-The title of the manuscript should be changed into "Detection and simultaneous differentiation of three-co-infected viruses in Zanthoxylum armatum, this because the main purpose of the manuscript is the detection.

-In figure 1 line 94,  the authors obtained three different amplicons with three different molecular sizes, why they did not sequenced them to be sure that they matched with viruses sequences wich lised on GeneBank.

-In figure 2  there is a band in molecular size about 230bp, this band are common between the three studied viruses, why they did not sequenced it either or used it in real time PCR for studying the virus propagation in the infected tissues.

-in materials and methods, the authors extracted RNA and they did not mentioned how they examined its purity and which apparatus they used for that (spectrophotometer, Nanodrop, etc.

-In figure 3, it appears that the sensitivity of the multiples RT-PCR was high than the normal specific RT-PCR, the question is, it is logic??.

Reviewer 2 Report

I strongly suggest to the authors to show a photo of Green Sichuan pepper plant with FYD symptoms. The best would be to show plants with and without mixed infection if possible, compared of course with an healthy plant.

The paper is well written and merit to be published

Reviewer 3 Report

The paper describes a new molecular method for the simultaneous diagnosis of 3  viruses of Z. armatum. The methodologies are not original, even if well executed. The new diagnostic test lacks some important parameters for its validation such as the analytical specificity, i.e. verifying whether the new primers also recognize different isolates of the same 3 viruses (inclusivity) and above all they do not recognize other non-target viruses of the same plant species (exclusivity ). Authors are advised to calculate this parameter. Moreover, authors should better detail the materials and methods of their research.  With the suggested modifications, the manuscript may be published with minor revisions.
